# Siamese Neural Networks on the Trail of Similarity in Bugs in 5G Mobile Network Base Stations

Sebastian Zarębski [1,2,*], Aleksander Kuzmich [1,*], Sebastian Sitko [1], Krzysztof Rusek [2] and Piotr Chołda [2]

1 NOKIA, Bobrzyńskiego 46, 30-348 Kraków, Poland
2 Institute of Telecommunications, AGH University of Science and Technology, Al. Mickiewicza 30, 30-059 Kraków, Poland
* Correspondence: sebastian.zarebski@nokia.com (S.Z.); aleksander.kuzmich@nokia.com (A.K.)

**Abstract:** To improve the R&D process by reducing duplicated bug tickets, we used the idea of composing a BERT encoder as a Siamese network to create a system for finding similar existing tickets. We proposed several different methods of generating artificial ticket pairs to augment the training set. Two phases of training were conducted. The first showed that only approximately 9% of pairs were correctly identified as certainly similar. Only 48% of the test samples were found to be pairs of similar tickets. With fine-tuning, we improved that result to 81%, which is a number describing a set of common decisions between the engineer in the company and the solution presented. With this tool, engineers in the company receive a specialized instrument with the ability to evaluate tickets related to a security bug at a level close to an experienced company employee. Therefore, we propose a new engineering application in corporate practice in a very important area with Siamese network methods that are widely known and recognized for their efficiency.

**Keywords:** fault detection; machine learning; natural language processing

## 1. Introduction

Companies like NOKIA are developing new technologies and their R&D process is inherently complex and requires significant financial and human resources. Employees of telecommunication companies involved in the implementation of new technologies should be aware of the current technical problems that other engineers face. To improve the process, it is necessary to design and create a system to search and compare newly discovered defects with those already fixed.

However, the bug descriptions related to the security of the 5G mobile network base stations are made up of many complementary fragments. In the NOKIA process, a title and a description are used to describe the symptom. It includes a list of the steps that led to the observation and a characteristic of the expected and observed test results. The noticed symptoms are also described by a set of system logs, automatically recorded during device operation. Bug descriptions (tickets) are made up of text fragments that are characterized by a very high accumulation of phrases and words that are not included in the general English dictionary. Thus, it is futile to expect that the application of neural networks, which were evaluated on open and general data sources, will produce similar results when used in the technological process. The specific corporate dataset is characterized by a high content of abbreviations and specific telecommunication and company jargon. It makes tickets difficult to understand without having the proper knowledge; see Figure 1. It presents a visualization of the error descriptions so that one can see how they resemble each other. Many of them intermingle or overlap, suggesting that the words they are made of are common to all classes. What is important is that security problems are found in different functional areas – there is no single area related to it. However, when looking at the figure, one can focus on classes fifth and sixth. These contain most of the bugs related to security. They blend with the content of errors related to other areas.

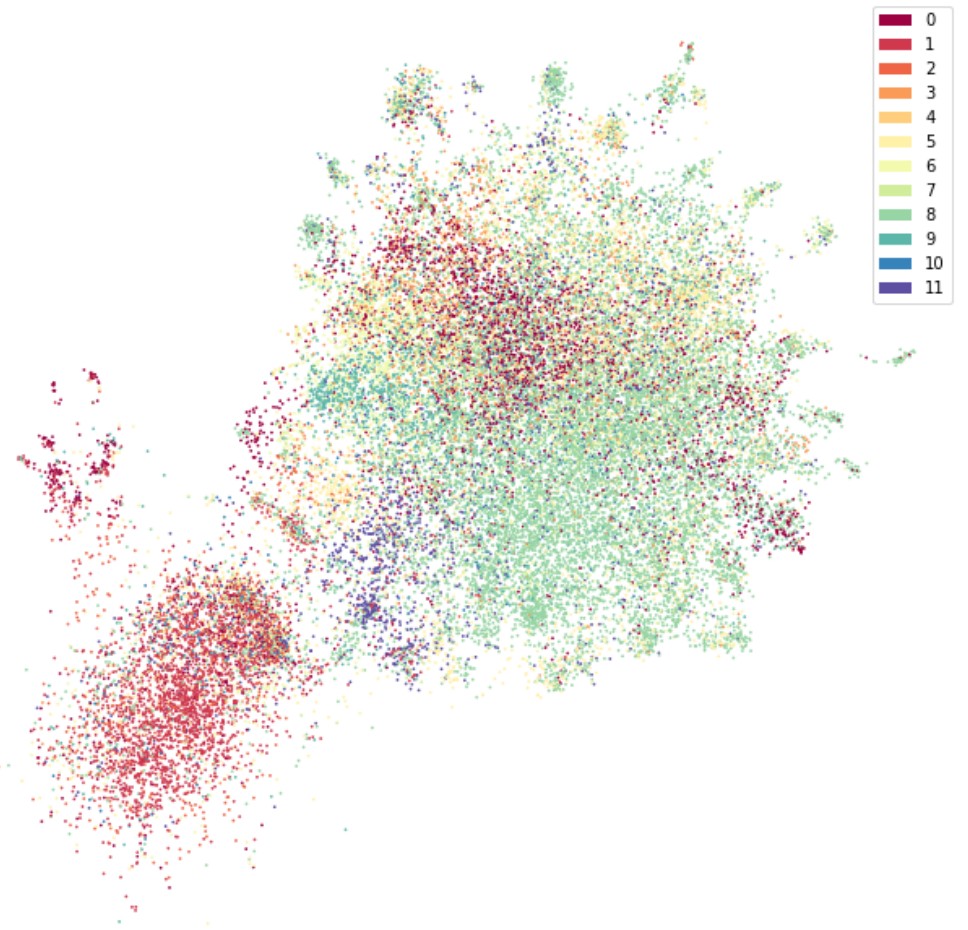

**Figure 1.** Visualization of the encoding of bug descriptions in 12 functional areas. Tickets form a large and rather uniform cluster, showing how similar they are. Algorithm used—UMAP [1] ($n_{\text{neighbors}} = 100$, $min\_dist - 0.5$).

In our work, we focus on using an existing neural network architecture (transformer) to create a system that finds semantically similar error descriptions. This was accomplished by using a combination of identical networks to make a so-called Siamese network. By applying additional filters in the form of categorical data (product name, software version, hardware type used), we improved the results by rejecting bug reports that were deceptively similar to each other. By creating a corpus of important vocabulary from 3GPP documentation [2], we were able to improve the efficiency of the overall tool and improve the processing of a text. With this, we improve the capabilities of engineers to detect similar security breaches in the code.

We give the following structure to this paper. In Section 2, we present the exact statement of the problem. In Section 3, we point to the literature that inspired our idea. In Section 4, we describe the methods we used. In Section 5, we present our results and findings. In Section 6, we discuss the findings in relation to the future work we plan to carry out. In Section 7, we conclude our research.

## 2. Problem Statement

In Nokia, as soon as the bug report is written, the fault coordinator reviews it to see if the problem is already known or if a similar one has already been submitted. If so, the ticket may be closed and declared invalid. In other cases, a so-called investigation phase begins, where a selected team of programmers and/or specification architects looks for the source of the problem. Then a fix is created and provided to a tester. If the patch is sufficient and the problem no longer exists, the ticket can be closed. This process is described in Figure 2.

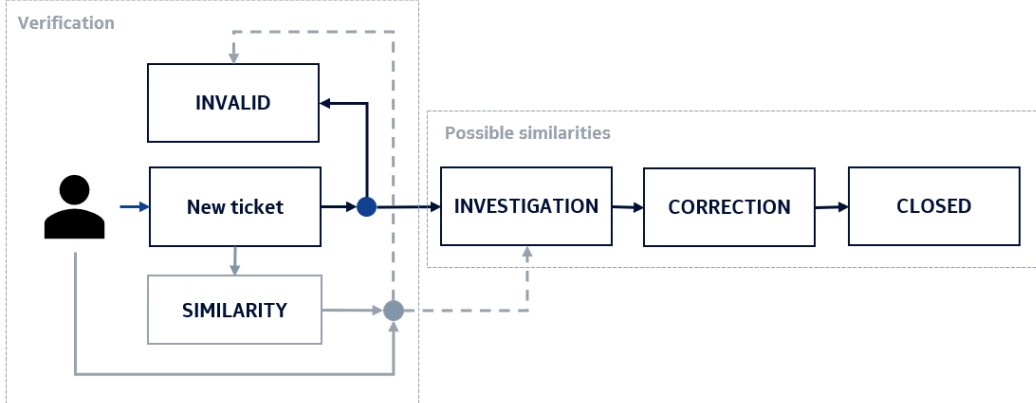

**Figure 2.** Typical bug report status diagram. The thick black line describes the present process, the thick grey line the proposed improvement of the process.

The company's software is developed using tools and libraries created and developed by other companies. If security-related errors are found, it is necessary to quickly fix them. Such an error may concern either a single product (e.g., a version of a base station) or their entire line (e.g., an error in the Open SLL library—CVE-2022-2274—that in special circumstances, an attacker may be able to trigger a remote code execution on the machine performing the computation) [3].

There is the possibility that testers from different offices may find the exact same error and create duplicate tickets. In the current process, there is a risk that two or more coordinators—especially from different offices and living in different time zones—will not notice that another department saw the same or similar error. Such a situation creates a strong need to enable such checking already in the phase of creating a ticket. It would improve the process by reducing the effort of coordinators and other people who can focus on different aspects of the project. Furthermore, such an improvement can lead to a situation in which all products and their versions (branches of a code) are fixed under the very same ticket.

Taking into account the content of the reported problems, some of them are described in a similar manner. It is characterized by an accumulation of IDs of problems found in software from other sources than internal:

**Ticket A:**
```
Info-Zip Unzip ≤ 6.10, ≤ 6.1c22 - Multiple Vulnerabilities
CVE-2018-1000035, CVE-2018-1000031, CVE-2018-1000032 and more
```
**Ticket B:**
```
Info-Zip Zip ≤ 3.0, Unzip ≤ 6.0 - Multiple Vulnerabilities -
CVE-2022-0530 and more
```

Tickets A and B are examples that might describe similar security issues (and, in fact, these were marked as such by a fault coordinator). One can see several parts of the text that start with a CVE prefix. The problem is that tickets having nearly identical syntax may be easily regarded as similar because they have the same text parts. On the other hand, it happens that two tickets marked by a fault coordinator as related to the same security problems have different content (yet, after a deep reading of their descriptions, one can find the reason for grouping such tickets). Titles of tickets C and D are examples of such.

**Ticket C:**
```
Mozilla Network Security Services (NSS) ≤ 3.68, ≤ 3.72 - Remote
Code Execution Vulnerability - 3.68.1, 3.73
```
**Ticket D:**
```
Python Package:  lxml -lt 4.6.5 - Local Improper Input Validation
Vulnerability - GHSA-55x5-fj6c-h6m8
```

Thus, a system that automatically finds similarities in such texts must be able to find the context based on parts of the words using subword information.

## 3. Literature Review

There are already well-known approaches to efficient text encoding. FastText algorithm introduced the use of *n*-grams in a skip-gram model, where cosine similarity was used to evaluate the effectiveness of a solution [4]. Approaches that do not use machine learning were developed, focusing on indexing textual data [5]. However, our scenario requires the use of a method that can be adapted to very inclusive meanings in a dataset. Therefore, a powerful neural network architecture must be utilized to find and understand complex meanings properly. Lately, the encoding model called the BERT was presented, claiming the first place in many evaluation metrics [6]. Another research proved that it might be used in a semantic comparison of sentences. For this, an architecture called a Siamese network [7] must be used. Before the BERT was introduced as a part of Siamese architecture, several researchers used different and widely known approaches CNN, GRU, or LSTM) [8]. Moreover, it was found that a combination of local and global features produces satisfactory sentence representations that can be utilized [9]. Several researchers have shown that the use of a BERT encoder in this scenario produces high results in multiple tasks [10,11]. Researchers of [12] compared BERT with ALBERT (a lighter version of the original BERT) and found that both architectures are sufficient for sentence-pair regression tasks such as semantic textual similarity. Moreover, by adding an additional outer CNN sentence-embedding network can help in improving metrics.

The authors of [13] worked on a comparison between different calculation methods to find text similarity using several Siamese networks. The experiment result shows that the Siamese network based on the BERT model with the Cosine Similarity metric achieved the highest accuracy rate. Based on that, we concluded to choose the Cosine Distance metric among others, like Pearson correlation coefficient, Spearman's rank correlation coefficient or simple Mean-square error.

As the authors of [14] stated, the Siamese network is good in calculating a semantic text similarity but lacks the interaction between the two sentences in the encoding process. Both sentences are treated as independent in the encoding process. Mentioned research proposes a method called the SENB that combines the Siamese network and the interaction model (ELECTRA [15]). The evaluation was done on the TwitterPPDB dataset and multiple different architectures were used, CNN, LSTM, BiGRU, and Transformer (standalone). The BERT and proposed SENB network achieved the best results, which proves that the Siamese network structure is better than the single traditional model in semantic similarity.

For sentence similarity tasks, two main loss functions are widely used, these are triplet loss and contrastive loss functions. Based on the known research, the former one tends to produce better overall results [16]. However, the mentioned research did not use BERT as a text encoder, which potentially would affect the final result. The problem of the triplet loss is that we need to manually create a set of three inputs per sample, making the general task even more difficult. There are situations in which it is hard to determine whether one sample is closer to the baseline sample than the other sample. Unlike the triplet, the contrastive loss function only takes two inputs: positive sample and negative example. This is why we concluded to use it in our approach, to make the whole solution as simple as possible.

Noticeably, training is required in two phases, normal, in which encoder weights are frozen (non-trainable), and second, called fine-tuning [17], in which we unlock the chosen number of encoder layers. To use the model, after encoding text into its dense representation, such vector must be checked against a database of existing bugs. For this, the Annoy library that extends the idea of the ANN (here: Approximate Nearest Neighbor) [18], may be used. It provides an optimal way of computing distances between pairs of vectors (Euclidean, cosine, etc.).

## 4. Methods

Our proposed methodology consists of two equally important steps. The first is the definition of the neural network model and a detailed explanation of its layers that make up the architecture of a Siamese neural network. In the second, we proposed a way to prepare a series of different datasets that describe the same problem but that will be able to describe different aspects of the behavior and performance of the proposed architecture against the technical dataset.

### 4.1. Model Used in the Research

Having prepared training and testing sets, we can determine the architecture of the neural network. Based on the research cited in Section Three, we focus on using two BERT encoders connected in a Siamese network, as presented in Figure 3.

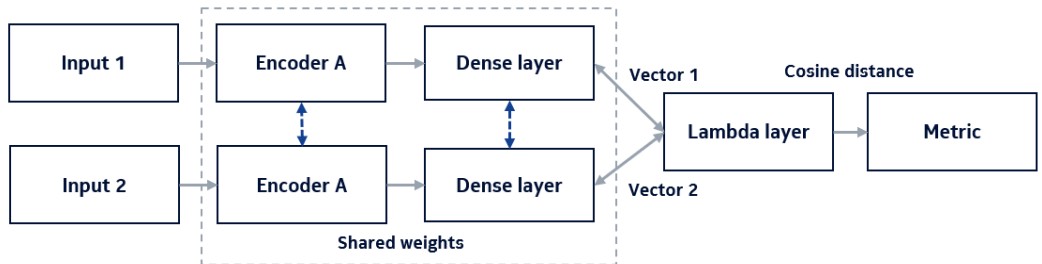

**Figure 3.** Generic Siamese network architecture with shared encoder and dense layer weights.

Because both encoders and dense layers share parameters (neurons' weights) with each other, this configuration can be simplified to the use of just one set of them.

For this research, the base uncased version of BERT was used (12 attention heads, 12 hidden layers, each with 768 neurons, that is, a version with 110 M parameters). We decided to use this model, although there are smaller versions of the model with reasonable performance metrics. We wanted to use the version with the largest number of available tokens that encode words related to the telecommunications domain and the memory space available to encode them. The model was composed of a single encoding block consisting of a BERT encoder followed by three layers—normalization layer, dropout layer with a rate equal to 0.1, and a dense layer with 64 neurons.

To compare a pair of tickets, two input layers are used. The encoder is terminated by a trainable deep layer, which learns the representation of vectors after they were pushed through it. The whole network is finished with a Lambda layer, whose purpose is to compute the cosine distance between the obtained encodings. In this process, we use a loss function called the **contrastive loss**. We omitted using a triplet loss considering the fact that we wanted to find out if such a network could learn differences just by the contrast of content of each sentence. The use of triplet loss is considered the next step of research. The intuition behind the contrastive loss is that we want our similar vectors to be as close to 1 as possible since $\log(1) = 0$ (that is, the loss equal to zero). However, since the network is computing cosine distance, not cosine similarity, the final labels are determined in such a way that $d = 0$ means that two vectors are ideally similar and $d = 1$ means that they are completely different. In the production scenario, the Lambda layer is omitted and dense vectors are fed to the ANN algorithm (here: Approximate Nearest Neighbor) to cluster the most similar ones. These are the final predictions.

As a result, during the first phase of the learning process, while the BERT layers were frozen (non-trainable), the part of the model that was trained had about 50,000 parameters.

We fine-tuned the whole model on a single NVIDIA GeForce RTX 2070 SUPER GPU card. During the training phase, we used a batch size of 32 and a learning rate of $1 \times 10^{-4}$. During the fine-tuning phase, we left the same batch size, and but reduced the learning rate to a value of $1 \times 10^{-5}$.

### 4.2. Dataset Preparation

The problem description is composed of several textual parts. These are the title, expected observation outcome, observed situation, hardware details, and set of logs pointing to the possible root cause of a symptom. To represent it as a data sample, we merge all parts into a single sentence. The text is then tokenized. Each token, which is rarely a whole word, is called a word piece, as mentioned in [6]. In our case, a final vector usually has between three and five hundred tokens. We have found that only 2% of the entire history of software bugs are known as very similar items (duplicates). Such are defined by the description of symptoms indicating the same bug, if detected in the same product with the same software version, on the same hardware configuration. The resulting subsets usually contain 2 to 5 reports linked to each other. One ticket is defined as a parent, and the rest are attached to it. The problem is visualized in Figure 4.

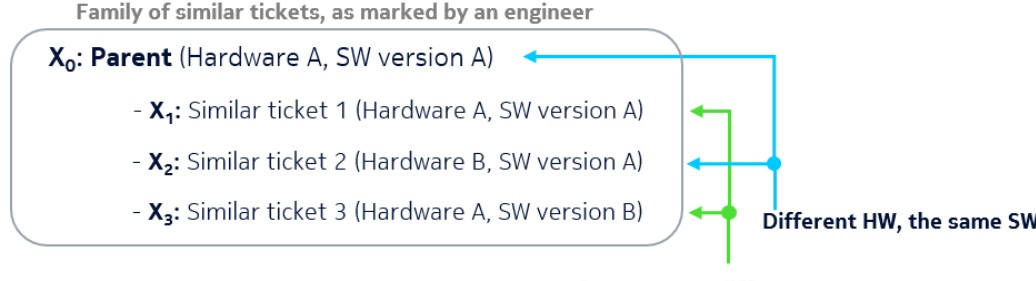

**Figure 4.** The explanation of the different types of similarity between tickets is marked as such by an engineer. The parent ticket $X_0$ is at the same time similar to $X_1$ (by having the same categorical values) and to ticket $X_2$ even if the following was found on different hardware.

The relatively small number of similar tickets is a positive fact for the company, but not for our problem. Ironically, by proposing this solution, we aim to reduce the small corporate dataset even further. For this reason, the final dataset must be specially prepared to compensate for a deficiency of training samples. Therefore, we need to artificially generate a larger number of pairs, which will allow the neural network to learn the relationships between selected text fragments. To achieve this goal, we define six types of ticket pairs, which we then extracted from the initial dataset and combined into separate datasets. The rules of creating them are as follows:

- Set A—tickets are combined only from the internal subgroup (related to the parent ticket, but without it),
- Set B— as in set A, but along with a parent ticket,
- Set C—tickets are combined between different subgroups (having different parent tickets, without those parent tickets),
- Set D—as in set C, but limited to the same hardware and software release version,
- Set E—tickets are randomly combined between parent tickets and those that did not have similar tickets (we are sure these are different bugs),
- Set F—tickets are combined between different subgroups (having different parent tickets).

Sets A and B are composed of pairs with a positive similarity relation, that is, pairs of tickets that are similar to each other. Sets C through F are sets of pairs of tickets that are different from each other. The test set is the same for all subsets and contains unique pairs of similar tickets (created analogously to set B). Thus, we can verify which approach produces better results and whether the network should specialize in determining the similarity or dissimilarity of the given inputs. There is an important fact to notice: sets E and F are created for validation purposes to check whether such neural networks actually learn anything and are able to distinguish between different tickets grouped in semi-random or manually controlled process.

In the result, each version of the dataset was approximately three to four times larger than the original. The next step is to divide them into training, validation, and testing subsets. By doing it naively (taking random samples up to a defined amount), we could eventually lead to a data leak (pairs generated from the same problem could be split between training and testing sets). To avoid such a situation, we propose the following short procedure:

1. The dataset must be divided into smaller groups (grouped by parent problem)—groups (or clusters) must be treated as "whole samples",
2. clusters must be sorted by the date of correction of parents (the latest problems at the top, to put most of them in the testing or validation subset—to evaluate the model against the current company's technological backlog),
3. then clusters may be separated into desired subsets.

The initial dataset was divided as follows—10% of the most recent samples describing pairs of tickets (in terms of when they occurred) of each product were selected for the test set. The pairs in this set were selected as in proposal B, because of their form resembled the reality (that is, because they were marked by the testers). In this way, we reconstructed a dataset that describes the latest trends in the company's software development process and is the same as the decisions of engineers in the company. Hence, this dataset was not altered or expanded in any way and was used to examine the training results for each dataset from A to F. From the remaining portion of the original set, an additional 20% of the most recent pairs were extracted and a validation set was created from them—these samples were selected as in the testing set. From the rest of the samples, successive versions of training sets A through F were created. The process of dataset preparation and models training is presented in Figure 5.

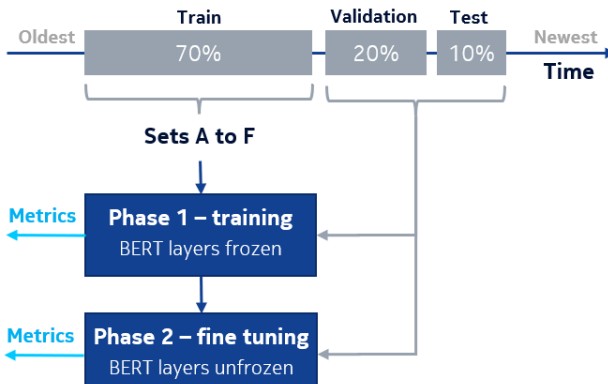

**Figure 5.** The visualization of the process of splitting training dataset into several variations and two phases of neural network training.

## 5. Results

Let us define our expectations for the results. Tables 1 and 2 contain the metrics of testing each set for the process with frozen encoder weights and unfrozen (fine-tuning). The first two columns show the percentage of pairs classified as similar when the neural network was very confident (distance $d = [0; 0.25]$) and confident ($d = (0.25; 0.5]$). The third column is the sum of these results, which can be compared to the fourth column, which contains the rest of the prediction results. The distance $d = (0.5; 1]$ means that the network considered these pairs to be rather dissimilar or completely different from each other.

The first phase of training, which assumes that the BERT encoder layers are frozen, proves that the internal vocabulary related to the problem is difficult to use, and the network is unable to capture much of the important context. The results are presented in Table 1. It is noticeable that the confidence of the answers in the similarity of the samples is quite low. At best, 9% strong confidence is obtained and only 48% of the test pairs are marked as

somewhat similar. These results are observed for set E, which represented pairs that were dissimilar to each other and that were selected almost randomly. An additional problem arises here because the pairs are combined almost independently of the company's fault management process. Due to randomness, there is no control over exactly which pairs would be formed. Thus, a different draw might change this result. Therefore, it is worth looking at the second-best result, that is, for set F, in which we combine dissimilar pairs by pairing tickets belonging to other groups (having different parent tickets). Interestingly, for the model with a frozen encoder, it was much easier to learn differences between samples set as such by fault coordinators. However, the results for similar pairs (marked in the same way) are much worse. It makes sense as in this approach, we create a dataset from carefully selected different tickets, in which most of the words and their context are easier to distinguish.

**Table 1.** A demonstration of the effectiveness of the neural network depending on the approach taken to create the training set—frozen encoder.

| Metrics | [0; 0.25) | (0.25; 0.5] | [0; 0.5] | (0.5; 1] |
|---------|-----------|-------------|----------|----------|
| SET A | 6.70% | 29.82% | 36.52% | 63.48% |
| SET B | 5.56% | 35.84% | 41.40% | 58.60% |
| SET C | 3.56% | 30.36% | 33.92% | 66.08% |
| SET D | 4.32% | 35.02% | 39.34% | 60.66% |
| SET E | 8.76% | 48.02% | 56.78% | 43.22% |
| SET F | 6.50% | 41.94% | 48.44% | 51.56% |

The great difference from these results can be observed in Table 2. It shows how crucial the phase of fine-tuning the network with the BERT encoder is. Our research has shown that in the case of smaller sets, such as in this case, it is enough to unlock only one encoder layer, that is, the last one, to obtain good results and save some memory on the GPU card. After training, the neural network was able to identify more than 81% of the test samples given as similar, of which only half a percent was probably similar. Only 17% of the samples could not be classified as related and similar.

**Table 2.** A demonstration of the effectiveness of the neural network depending on the approach taken to create the training set—fine tuning.

| Metrics | [0; 0.25) | (0.25; 0.5] | [0; 0.5] | (0.5; 1] |
|---------|-----------|-------------|----------|----------|
| SET A | 81.12% | 0.56% | 82.68% | 17.32% |
| SET B | 80.34% | 0.50% | 80.84% | 19.16% |
| SET C | 79.92% | 0.82% | 80.74% | 19.26% |
| SET D | 76.82% | 0.84% | 77.66% | 22.34% |
| SET E | 75.40% | 0.82% | 76.22% | 23.78% |
| SET F | 79.54% | 0.54% | 80.08% | 19.92% |

By enabling the encoder layer to be trainable, making it possible to learn the actual distribution of the tokens in the datasets, we found that such a neural network can learn the context in a way that is able to find similarities between ticket pairs. It is visible when comparing the results for sets A and B (similarities) with the rest (differences); we reverse the situation from previous measurements. The network turned out to be powerful enough to catch small differences in the given pairs of texts and correctly find connections between them. This is especially needed for security-related tickets that consist of several external bug identifiers (like the previously mentioned CVE errors) or important keywords.

## 6. Discussion

The dataset we have come to work with is internal to the company, and moreover, the proposed method relies heavily on its further transformations. Thus, there are no

"common" metrics with other studies (other than general ones like accuracy), but it is impossible to relate them to other studies due to the different datasets used in the research. The main KPI we used was to compare the results of the model evaluation with the same decisions made by an engineer in the company (human assessment). Depending on the prepared dataset, that is, based on the interpretation of the data and the problem itself, the same network is able to evaluate 75% to 81% of the samples identically. That is, the main metric is to compare the automated system's evaluation of the similarity of two samples with how they were evaluated by engineers working in the company.

One criticism is that the BERT encoder model is unnecessarily large if one comes to using such a model in an internal production environment. The tokenizer that it uses was trained in different datasets and its tokens corpus is not the best in terms of describing all the meanings in the corporate, industrial dataset. It should be noted that in the process of creating 5G network base stations, Nokia prepared about 500 of its hardware versions (motherboards and radio heads). Based on the document prepared by the 3GPP organization that contains all abbreviations and phrases, we were able to identify approximately 3500 technical abbreviations and a dictionary containing approximately 30,000 words (including 40% not found in the common English dictionary). Many of them are crucial to understanding the meaning of text related to security issues. As technology and backlog are constantly changing, meanings (and vocabulary) must be updated. At some point, the corpus of tokens used by the BERT might lose its ability to properly split technical texts into meaningful tokens. Due to its technical requirements and demand for resources (enormous size of the dataset to converge during a training—regarding one available in the companies as NOKIA or multiple GPU cards required), it foreshadows itself as only a temporary solution.

Despite designing and evaluating a number of datasets, the procedure for generating a test set is still to be improved. It is still naive from a business perspective, as the algorithm does not pay any attention if samples of each product (e.g., radio head types) are represented equally in the training and testing sets. This can be improved. Moreover, we did not measure the false positive rates yet. This research is just a first step in creating an automated way to find a similar meaning from the business and technical perspectives. The next and very important step we will approach is limiting the size of possible false similarities by applying filters on the so-called categorical values. These are related to the tickets and describe different (and additional) features of the product in which the bug is found. Examples of such are software version (branch name), connected hardware peripherals, testline version, code features under testing, and many different. This is supposed to help in situations as described below.

> **Ticket G:**
> `SW download failed when IPsec for eCPRI management plane is enabled`

There was a situation where several problems related to the same root cause were described in different tickets. The problem was introduced internally in the code itself as due to configuration misalignment, and on some rare occasions the TCP tunnel would not be set properly. This was clearly a security issue, disabling the gNB station's possibility to upgrade its software to a newer, improved version. However, each ticket with the same root cause could be considered separate (from a business perspective), since those were created for different products and software versions, sometimes having a few months of a time difference between them. A categorical value filter may improve the general accuracy in such cases.

An important aspect not addressed in this work is the uncertainty of the response. For data with such specific content, it would be useful to be able to estimate the uncertainty in assessing the similarity of two tickets. Then, we should consider applying the problem in the context of variational inference. This will allow the solution to correctly estimate two types of uncertainty, epistemic (related to lack of knowledge) and aleatoric (related to

probability calculus itself). If we want to suggest potentially similar error descriptions to the user, we must also be able to answer how confident we are in giving such an answer.

## 7. Conclusions

Eventually, we obtained a highly automated system that can be used to support the daily work of the testers in the company. However, it should be noted that this study is based on working with pairs that we know are similar. For this purpose, we used data provided by the company's engineers. Their decisions, assessing which errors are similar to each other, were used to create the datasets used in this research. Despite the identified weaknesses of the solution, we discovered that the approach we presented was able to describe 80% of all tickets the same way as the company's employees.

**Author Contributions:** Conceptualization, S.Z.; methodology, S.Z.; software, S.S. and A.K.; data curation, S.S. and A.K.; formal analysis: S.Z., K.R., and P.C.; writing—original draft preparation, S.Z.; writing—review and editing, S.Z., K.R., and P.C.; supervision, S.Z., K.R., and P.C. All authors have read and agreed to the published version of the manuscript.

**Funding:** This research received no external funding.

**Data Availability Statement:** Not applicable. Data are related to the internal processes of the NOKIA company.

**Conflicts of Interest:** The authors declare no conflict of interest.

## Abbreviations

The following abbreviations are used in this manuscript:

| | |
|---|---|
| R&D | Research and development |
| BERT | Bidirectional Encoder Representations from Transformers |
| 3GPP | The 3rd Generation Partnership Project |
| ANN | Approximate Nearest Neighbor |
| GPU | Graphics processing unit |

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
