# Peer review of "Siamese Neural Networks on the Trail of Similarity in Bugs in 5G Mobile Network Base Stations"

_electronics, doi:10.3390/electronics11223664_

Round 1

Reviewer 1 Report

In general, the article describes the process of recognizing bugs descriptions that are close in meaning. Compared texts have specific vocabulary, which obviously makes it difficult to solve the problem.

The article is generally written clearly and is devoted to an important industrial problem. The unresolved problems are well described in the conclusion. However, several serious shortcomings make it difficult to understand the proposed method and its possible application in other studies.

Major remarks

1)      The Literature review section is poor. It does not describe the state of the problem and how to solve it. As a result, it is difficult for the reader to understand what is new in this study. Meanwhile, a simple search yields a few links to relevant articles. For example

Saedi, C., & Dras, M. (2021). Siamese networks for large-scale author identification. Computer Speech & Language70, 101241.

Zhu, Y., Kuang, L., & Chen, Z. (2021, August). Siamese ELECTRA Network Combined with BERT for Semantic Similarity. In 2021 16th International Conference on Computer Science & Education (ICCSE) (pp. 481-485). IEEE.

Renjit, S., & Idicula, S. M. (2021, September). Siamese Networks for Inference in Malayalam Language Texts. In Proceedings of the International Conference on Recent Advances in Natural Language Processing (RANLP 2021) (pp. 1167-1173).

Han, S., Shi, L., Richie, R., & Tsui, F. R. (2022). Building Siamese Attention-Augmented Recurrent Convolutional Neural Networks for Document Similarity Scoring. Information Sciences.

Wang, K., Zeng, Y., Meng, F., & Yang, L. (2021, July). Comparison between Calculation Methods for Semantic Text Similarity based on Siamese Networks. In 2021 4th International Conference on Data Science and Information Technology (pp. 389-395).

2)      The computer vision problem is often solved using Siamese networks [Taigman, Y.; Yang, M.; Ranzato, M.A.; Wolf, L. Deepface: Closing the gap to human-level performance in face verification. In Proceedings of the Proceedings of the IEEE conference on computer vision and pattern recognition, 2014; pp. 1701-1708.], when two images are processed by two identical pre-trained networks. The obtained results (image vectors) are compared using a triplet loss function, which can be implemented as a triplet embedding distance [Schroff, F.; Kalenichenko, D.; Philbin, J. Facenet: A unified embedding for face recognition and clustering. In Proceedings of the Proceedings of the IEEE conference on computer vision and pattern recognition, 2015; pp. 815-823.] or a triplet of probabilistic embeddings [Sankaranarayanan, S.; Alavi, A.; Castillo, C.D.; Chellappa, R. Triplet probabilistic embedding for face verification and clustering. In Proceedings of the 2016 IEEE 8th international conference on biometrics theory, applications and systems (BTAS), 2016; pp. 1-8.]. What type of triplet loss function is used in the proposed method? What does lambda layer mean?

3)      The training process is not clear. Please describe the training and testing process in a little more detail.

Minor remarks

4)      Figure 1 contains minor typos. I think different vectors are at the input.

5)      Abbreviation ANN is often used in the sense of an artificial neural network.

Author Response

Hello,

Thank you for your feedback and suggestions!

Please find out answers and comments in the attached message. Additionally, please find the latexdiff for changes we introduced after your review (file: diff_after_revision_1.text).

Regards,

Sebastian

Reviewer 2 Report

1.     Expends the introduction to identify the problem. Weather the problem exists in the recent literature, cite the references in the introduction.

2.     Why the problem presented in the paper is challenging and why the existing solutions are not adequate to address this problem?

3.     The literature review is limited. Review more recent works and include in the text.

4.     Write a separate section of the Conclusion.

Author Response

(The authors gave the same response as above.)

Reviewer 3 Report

Dear authors, 

Although your proposal is interesting, you need to provide more info to make your paper suitable for replication and therefore publication. 

1. Provide more specific details about neural network model employed.

2. Provide info about the hardware and software implementation of your work.

3. Provide details about the datasets for training, validation and testing: features, number of samples per class, etc.

4. Provide the validation methods for your paper.

5. Provide the performance measures for your paper. 

6. Compare your results against other state-of-art methods. 

Author Response

(The authors gave the same response as above.)

Round 2

Reviewer 1 Report

The authors corrected the text of the article in accordance with the comments. In my opinion, the article can be accepted in the present form.

Reviewer 2 Report

Accept

Reviewer 3 Report

Dear authors,

I finished the review of your paper. Thank you very much for addressing my recommendations, I have no further on this work.